# Incidence of respiratory distress and its predictors among neonates admitted to the neonatal intensive care unit, Black Lion Specialized Hospital, Addis Ababa, Ethiopia

**Yared Asmare Aynalem**[1]*, **Hussien Mekonen**[2©], **Tadesse Yirga Akalu**[3], **Tesfa Dejenie Habtewold**[1,4‡], **Aklilu Endalamaw**[5‡], **Pammla Margaret Petrucka**[6‡], **Wondimeneh Shibabaw Shiferaw**[1©]

1 College of Health Science, Debre Berhan University, Debre Berhan, Ethiopia, 2 College of Health Sciences, School of Nursing and Midwifery, Addis Ababa University, Addis Ababa, Ethiopia, 3 College of Health Science, Debre Markos University, Debre Markos, Ethiopia, 4 Department of Epidemiology, University Medical Centre Groningen, University of Groningen, Groningen, The Netherlands, 5 Department of Pediatrics and Child Health Nursing, School of Health Science, College of Medicine and Health Sciences, Bahir Dar University, Bahir Dar, Ethiopia, 6 College of Nursing, University of Saskatchewan, Saskatoon, Canada

© These authors contributed equally to this work.
‡ These authors also contributed equally to this work.
* yaredasmare123@gmail.com

**Data Availability Statement:** All datasets analyzed in this study are publicly available. We have

## Abstract

### Background

Although respiratory distress is one of the major causes of neonatal morbidity and mortality throughout the globe, it is a particularly serious concern for nations like Ethiopia that have significant resource limitations. Additionally, few studies have looked at neonatal respiratory distress and its predictors in developing countries, and thus we sought to investigate this issue in neonates who were admitted to the Neonatal Intensive Care Unit at Black Lion Specialized Hospital, Ethiopia.

### Methods

An institution-based retrospective follow-up study was conducted with 571 neonates from January 2013 to March 2018. Data were collected by reviewing patients' charts using a systematic sampling technique with a pretested checklist. The data was then entered using Epidata 4.2 and analyzed with STATA 14. Median time, Kaplan-Meier survival estimation curves, and log-rank tests were then computed. Bivariable and multivariable Gompertz parametric hazard models were fitted to detect the determinants of respiratory distress. The hazard ratio with a 95% confidence interval was subsequently calculated. Variables with reported p-values < 0.05 were considered statistically significant.

### Results

The proportion of neonates with respiratory distress among those admitted to the Black Lion Specialized Hospital neonatal intensive care unit was 42.9% (95%CI: 39.3–46.1%) The

uploaded the minimal anonymized data set necessary to replicate our study findings as a Supporting Information file.

**Funding:** The authors have also confirmed that no financial funding was received for the study, authorship, and publication of this article.

**Competing interests:** The authors have declared that no competing interests exist.

**Abbreviations:** APGAR, Appearance pulse grimace activity respiration; CI, confidence interval; COR, crude odds ratio; GA, gestational age; HIV, human immunodeficiency virus; HMD, hyaline membrane disease; HR, hazard ratio; NICU, neonatal intensive care unit; PNA, perinatal asphyxia; PROM, prolonged rupture of membrane; RD, respiratory distress.

incidence rate was 8.1/100 (95%CI: 7.3, 8.9). Significant predictors of respiratory distress in neonates included being male [Adjusted hazard ratio (HR): 2.4 (95%CI: 1.1, 3.1)], born via caesarean section [AHR: 1.9 (95%CI: 1.6, 2.3)], home delivery [AHR: 2.9 (95%CI: 1.5, 5,2)], maternal diabetes mellitus (AHR: 2.3 (95%CI: 1.4, 3.6)), preterm birth [AHR: 2.9 (95%CI: 1.6, 5.1)], and having an Apgar score of less than 7 [AHR: 3.1 (95%CI: 1.8, 5.0)].

## Conclusions

In this study, the proportion of respiratory distress (RD) was high. Preterm birth, delivery by caesarean section, Apgar score < 7, sepsis, maternal diabetes mellitus, and home delivery were all significant predictors of this condition. Based on our findings this would likely include encouraging more hospital births, better control of diabetes in pregnancy, improved neonatal resuscitation and addressing ways to decrease the need for frequent caesarean sections.

## Background

Respiratory distress (RD) is a common problem for newborns immediately following birth. It is often seen during the transition from fetal to neonatal life. RD typically manifests in newborns as tachypnea, intercostal retractions, nasal flaring, audible grunting, and cyanosis. The successful transition from fetal to neonatal life requires a series of rapid physiologic changes in the cardiorespiratory systems. These changes result in a redirection of gas exchange from the placenta to the lungs and requires the replacement of alveolar fluid with air and the onset of regular breathing [1]. Although RD may be transient in some newborns, if it persists, then there is a need for proper diagnostic and therapeutic interventions to optimize outcomes and minimize morbidity.

RD is one of the most common reasons for neonates to be admitted to the neonatal intensive care unit (NICU) [2, 3]. Fifteen percent of term infants and 29% of late preterm infants admitted to the NICU develop significant respiratory morbidity [4]. This incidence is even higher for infants born before 34 weeks' gestation [5]

Certain risk factors increase the likelihood of neonatal RD. Recognized causes of RD in other low and high resource countries includes; prematurity, low first and fifth minute Apgar scores, meconium aspiration syndrome, caesarian section delivery, gestational diabetes, maternal chorioamnionitis, premature rupture of membranes [6], and oligohydramnios, as well as structural lung abnormalities are some predictors identified in previous studies [5, 7–10]. Other common causes include transient tachypnea of the newborn, meconium aspiration syndrome, pneumonia, sepsis, pneumothorax, and persistent pulmonary hypertension of the newborn [11]. In contrast, the risk decreases with each advancing week of gestation and birth through spontaneous vaginal delivery [12].

Regardless of the cause, if not recognized and managed quickly, RD can escalate to respiratory failure, cardiopulmonary arrest, and even death. Therefore, it is imperative that any health care practitioner caring for newborn infants be able to readily recognize the signs and symptoms of RD, differentiate the various causes, and initiate management strategies to prevent significant complications or death. Consequently, neonates in need of critical medical attention are usually admitted to the NICU. These infants tend to be preterm, have a low birth weight, or have serious medical conditions including RD [13, 14].

Globally, there are different policies, strategies, and programs which work on or advocate for the prevention and care of preterm neonates and their birth outcomes, including RD, like the Sustainable Development Goals (SDGs) and the Every Women and Every Child initiative [15, 16]. Despite these efforts, RD remains among the leading causes of neonatal mortality and morbidity [17–21]. Indeed, in Ethiopia, RD is the most common cause of neonatal mortality and morbidity [17–22], resulting in exponentially increasing neonatal care costs within the first 28 day of life. Additionally, few studies have been conducted in developing countries to assess RD in these regions, including Ethiopia. Therefore, this study we aimed to determine the incidence and predictors of RD among neonates who were admitted to the NICU at Black Lion Specialized Hospital, Ethiopia.

## Methods

### Study design, setting, and population

An institution-based retrospective follow-up study was conducted among a cohort of neonates from the previous consecutive five years (from January 2013 to March 2018). The study was conducted in Addis Ababa, a capital city of Ethiopia at NICU of black lion hospital. Addis Ababa has ten sub-cities in which the City lies at an altitude of 7,546 feet (2,300metres). It has twelve governmental and nine nongovernmental hospitals. The NICU of black lion hospital ward is able to accommodate a maximum of 60 patients with average of 20–40 patients' daily admission. There are on average 5000–6000 annual admissions. The study was conducted from March to April 1, 2018. The neonatal chart number were taken from the HMIS- data base. The total patients admitted to NICU from January 2013-last of March 2018 were 5000. We have found the number of admissions for each year. The samples were proportionally allocated for each year, and with systematics sampling; the study participants of each year were selected as follows. First, numbering the units of each year on the frame from 1 to N (N = total admission of each year), then we determine the sampling interval (K) by dividing the number of units in the population by the desired sample size of each year (n = sample size of each year) which gives 8. Then number between one and 8 at random was selected (2 were selected). This number is called the random start and the first number included in the sample. Then later Selection was conducted every 8th unit after that first number. Our source population was all neonates admitted to the NICU at the Black Lion Specialized Hospital, Ethiopia. All neonates who were admitted to the NICU in the previous five consecutive years (from January 2013 to March 2018) were considered as the study population.

### Eligibility criteria

All targeted neonates' medical cards documented in the previous five years from the study period were recruited and those with incomplete cards were excluded.

### Sample size determination and sampling procedure

The sample size was determined via the double population proportion formula using Epi-Info Version 7 by assuming a one-to-one ratio of exposed to non-exposed, 95% level of confidence, and a power of 80%. We considered four significantly associated factors to calculate the sample size; the largest sample size was 522. After adding a 10% non-response rate, the total sample size became 604. The neonates' cards were accessed using the systematic random sampling technique after determining the sampling fraction (k = 6) and the first card was selected by the lottery method.

## Study variables

**Dependent variable.** Incidence of RD

**Independent variables.** *Socio-demographic factors.* Neonatal-related variables included age at admission, gestational age, sex, the weight of the neonate, date of NICU admission and discharge. Maternal-related variables were age and residency.

*Gynecologic-obstetric related factors.* Antenatal care (ANC) follow up, gravidity, parity, mode of delivery, multiple pregnancies, PROM, preeclampsia, abruption placenta, and breast-feeding initiation.

*Medical disorders in mother.* Hypertension, diabetes mellitus (DM), human immune virus/acquired immune deficiency syndrome (HIV)/(AIDS).

*Neonatal outcome condition.* Apgar score, sepsis, jaundice, hypothermia, prenatal asphyxia (PNA), hypoglycemia, meningitis, esophageal atresia.

*Data collection tools.* A pretested checklist was used to collect the required data from the neonates' charts. The checklist was translated to the local language of Amharic and back to English. The consistency of this translation was checked to ensure its accuracy. Data were collected by reviewing the complete patients' cards from the previous five consecutive years from the study period. RD was confirmed by reviewing neonate medical charts.

*Data quality control.* Data quality was assured by designing proper data abstraction tools. The checklist was evaluated by experienced researchers. The data collection instrument was pretested on 5% of the sample size. Rigorous training was given regarding the data abstraction checklist and data collection process for both data collectors and supervisors. During the data collection time, close supervision and monitoring were carried out by the supervisors and investigator. Double data entry was also done using Epi Data 4.2.0 software.

*Data processing, analysis, and presentation.* Before analysis, data was cleaned, edited, and coded. Any errors identified at this time were corrected after review of the original data using the code numbers that we had assigned during the data collection period. Data were entered using Epi-Data version 4.2.0 and analyzed using STATA 14 statistical software. Incidence density rate (IDR) was calculated for the entire study period. Subsequently, the number of cases of RD within the follow-up period was divided by the total person-time at risk on follow-up and reported per 100-person day. Kaplan-Meier survival curves were used to estimate the mean survival time and the log-rank tests were used to compare survival curves. Proportional hazard assumption was tested both graphically and through the Schoenfeld residual test for all predictors, revealing that the proportional hazard assumption was met. After checking this assumption, by comparing models, a more effective hazard model was selected using the log likelihood ratio (LR) test and the Akaike Information Criterion (AIC). In this parametric approach, the baseline hazard and the vector of its parameters were assessed together with the regression coefficients. The best-fit model was chosen using AIC; selecting those having the smallest AIC. Subsequently, parametric models were completed for neonates to ascertain the possible predictors. Variables having a p-value less than or equal to 0.05 in the bivariate analysis were fitted to the multivariable Gompertz hazard distribution regression model with a 95% confidence interval. A p-value less than 0.05 was considered statistically significant.

*Ethical consideration.* Ethical clearance was obtained from Addis Ababa University, College of health science ethical review board. Letters of cooperation were written to Black Lion Specialized Hospital by the ethical review board members and subsequently permission was obtained from the clinical director and relevant department and unit heads of the hospital. Since the study was conducted by taking appropriate information from medical chart, it will not inflict any harm on the patients. The name or any other identifying information was not

be recorded on the checklist and all information that was taken from the chart was kept strictly confidential and in a safe place.

Following these approvals, access to the medical charts was provided and we did our utmost to maintain participant confidentiality by storing in a file cabinet and kept in a key and locked system with computer pass ward.

### Operational definition

**Event (neonatal RD).** The presence of two or more of the following signs: an abnormal respiratory rate (tachypnea > 60 breaths/min, bradypnea < 30 breaths/minute, respiratory pauses, or apnea) or signs of labored breathing (expiratory grunting, nasal flaring, intercostal recessions, xyphoid recessions), with or without cyanosis.

**RD.** presence of two or more of the following signs: an abnormal respiratory rate, expiratory grunting, nasal flaring, chest wall recessions, and cyanosis as per patient chart information.

## Results

### Characteristics of neonates

Among 604 neonatal charts reviewed, 571 (94.5%) records met the enrollment criteria and were included in the final analysis. Of this group, 299 (52.34%) of the study participants were males. Neonates in the late neonatal period accounted for more than half of the study participants. The mean age of the cohort at the time of admission to the NICU was 3 days ± 3.72 standard deviation (SD). More than half of the neonates admitted to the NICU were diagnosed with neonatal sepsis, but other common causes of admission were jaundice, hypothermia, and PNA (Table 1). In addition, the common types of RD for neonatal admission were RDS or hyaline membrane diseases (Fig 1).

### Socio-demographic and obstetric characteristics of mothers

In the current study, most mothers were found to be between the ages of 20–34. The mean age of mothers was found to be 28 years ± 5.42 SD. Among all mothers enrolled in this study, 336 (58.9%) experienced spontaneous vaginal delivery. Regarding obstetric, gynecological, and

**Table 1. Characteristics of neonates admitted to the NICU at Black Lion Specialized Hospital, Ethiopia, (n = 571).**

| Characteristics | Category | Frequency (%) |
|---|---|---|
| Sex | Male | 299 (52.34) |
| | Female | 272 (47.66) |
| Gestational age (weeks) | <37 | 239 (41.8) |
| | ≥37 | 332 (58.2) |
| Neonates weight (g) | <2500 | 243 (42.6) |
| | ≥2500 | 328 (57.4) |
| Hypothermia | Yes | 180 (31.5) |
| | No | 391 (68.5) |
| Sepsis | Yes | 260 (45.5) |
| | No | 247(43.3) |
| Jaundice | Yes | 202 (35.4) |
| | No | 369 (64.6) |
| 1st minute Apgar | <7 | 337 (59.0) |
| | ≥7 | 234 (41) |

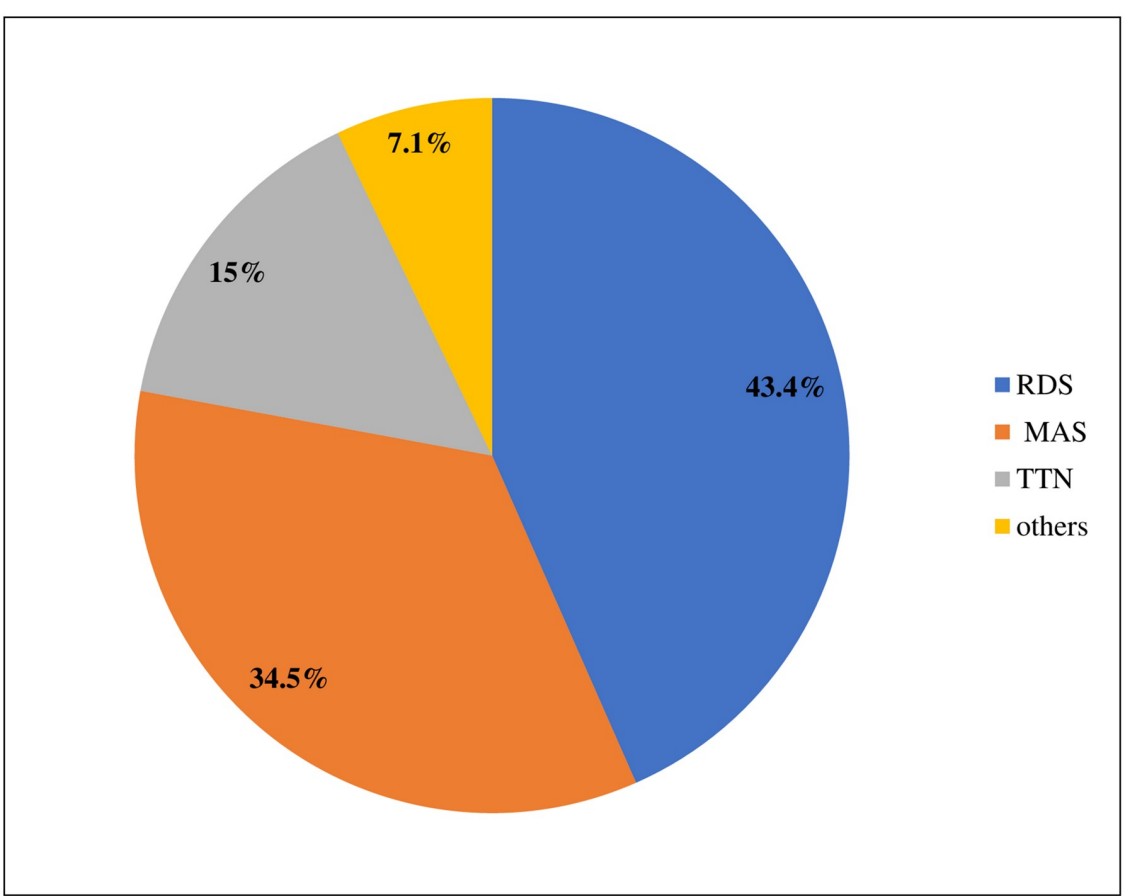

**Fig 1. Common types of neonatal RD at Black Lion Specialized Hospital, Ethiopia.**

medical diagnosis of maternal diseases, 250 (43.8%) had PROM, (43.8%) had HIV/AIDS (13.5%), and (10.7%) had DM. The results of this study also indicated that the majority [402 (70.4%)] of neonates were born to mothers who had an ANC follow-up. (Table 2).

## Overall proportion and incidence rate of RD in neonates

The overall proportion of neonates that develop RD was found to be 42.9% (95%CI: 39.3–46.1). The overall incidence rate of RD was found to be 8.1 per 100 neonate day (95%CI: 7.29, 8.9) with 4331-person day observation.

## Time to discharge of neonates with RD

The overall median length of hospital stay for neonates with RD in this study was 9 days (95% CI: 8–10) and the overall length of hospital stay were 28 neonates' days (IQR5, 30 neonate-days). The cumulative probability of neonates not developing RD at the end of the first day in the NICU was 94.4%, between the fifth and 10th days was 41.3%, and between 20–28 days in the NICU was 19.14% (Fig 2).

The proportional hazard assumption was evaluated using Kaplan-Meier survival curves and the Sheffield residual global test and was found to be met ($x^2$ = 5.11; p value = 0.08) (Fig 3).

**Table 2. Socio-demographic and obstetric characteristics of mothers of neonates admitted to the NICU at Black Lion Specialized Hospital, Ethiopia, (n = 571).**

| Characteristics | Category | Frequency (%) |
| --- | --- | --- |
| ANC follow-up | Yes | 402(70.4) |
| | No | 169(29.6) |
| Maternal age (years) | <20 | 61(10.9) |
| | 20–34 | 426 (74.6) |
| | >34 | 84 (14.7) |
| Place of delivery | Home | 194(34) |
| | Health institution | 377(66.0) |
| Multiple pregnancy | Yes | 49(8.5) |
| | No | 522(91.5) |
| PROM | Yes | 250(43.8) |
| | No | 321(56.2) |
| HIV/AIDS | Yes | 76(13.3) |
| | No | 495(86.7) |
| Maternal DM | Yes | 61(10.7) |
| | No | 510(89.3) |

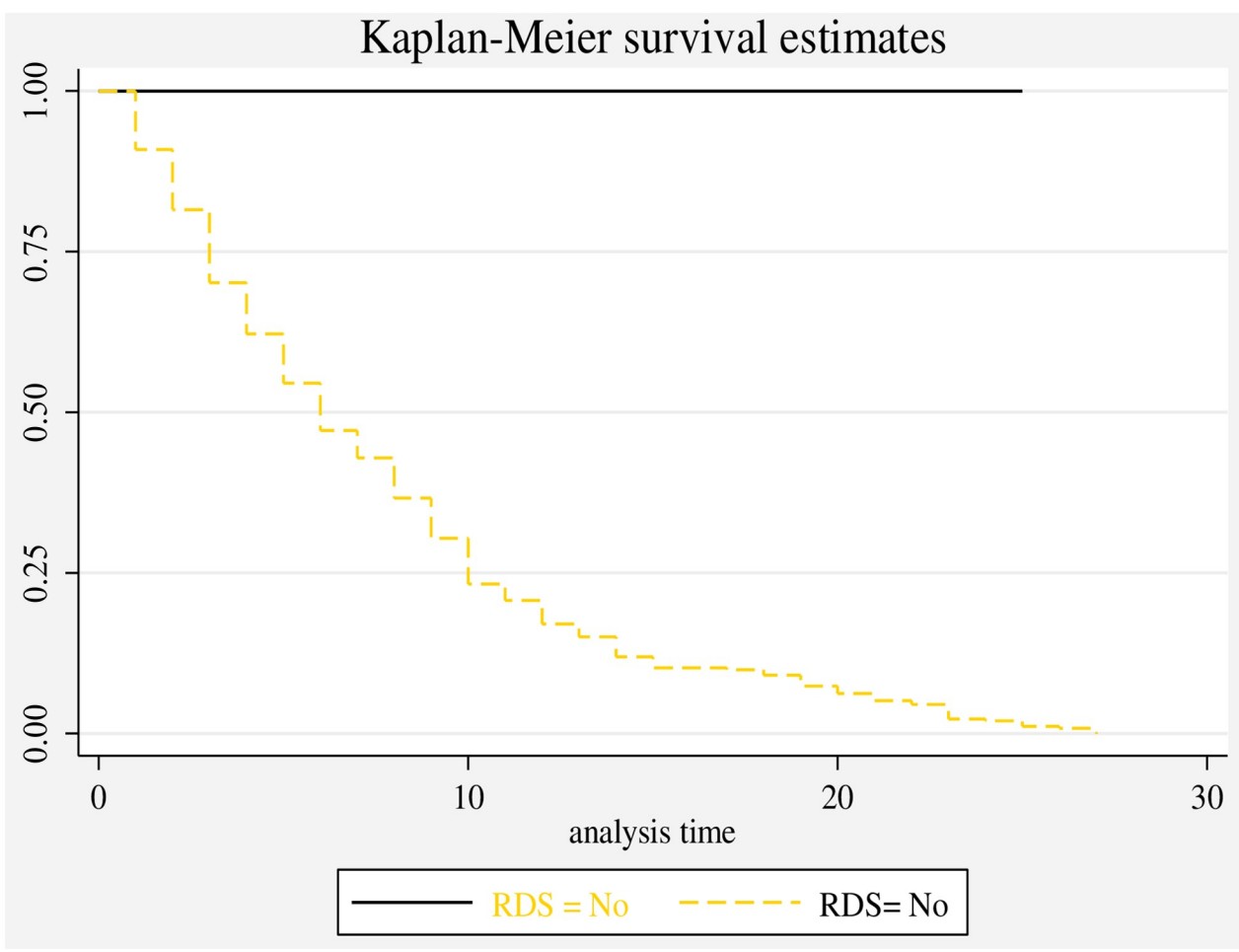

**Fig 2. Overall Kaplan-Meier survival estimate of neonates with RD admitted to the NICU at Black Lion Specialized Hospital, Ethiopia.**

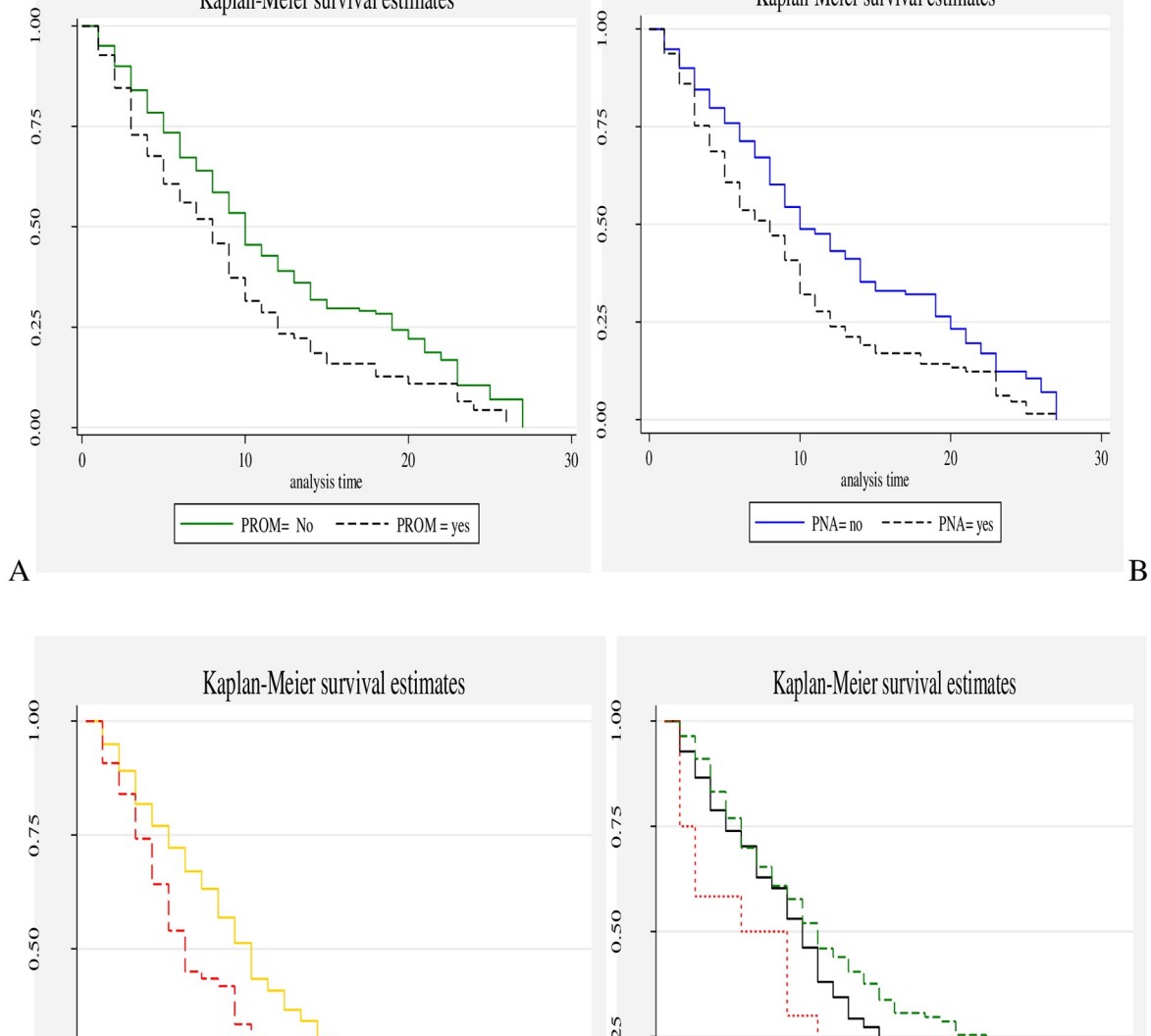

**Fig 3. Kaplan-Meier survival curves of neonates with RD with respect to A) PROM, B) PNA, C) maternal HIV/AIDS, and D) mode of delivery.**

## Model comparison criteria

The goodness of fit model was checked using the Cox-Snell residual test. Based on the AIC, the univariate Gompertz hazard distribution (AIC = 435.8) model was more efficient than the

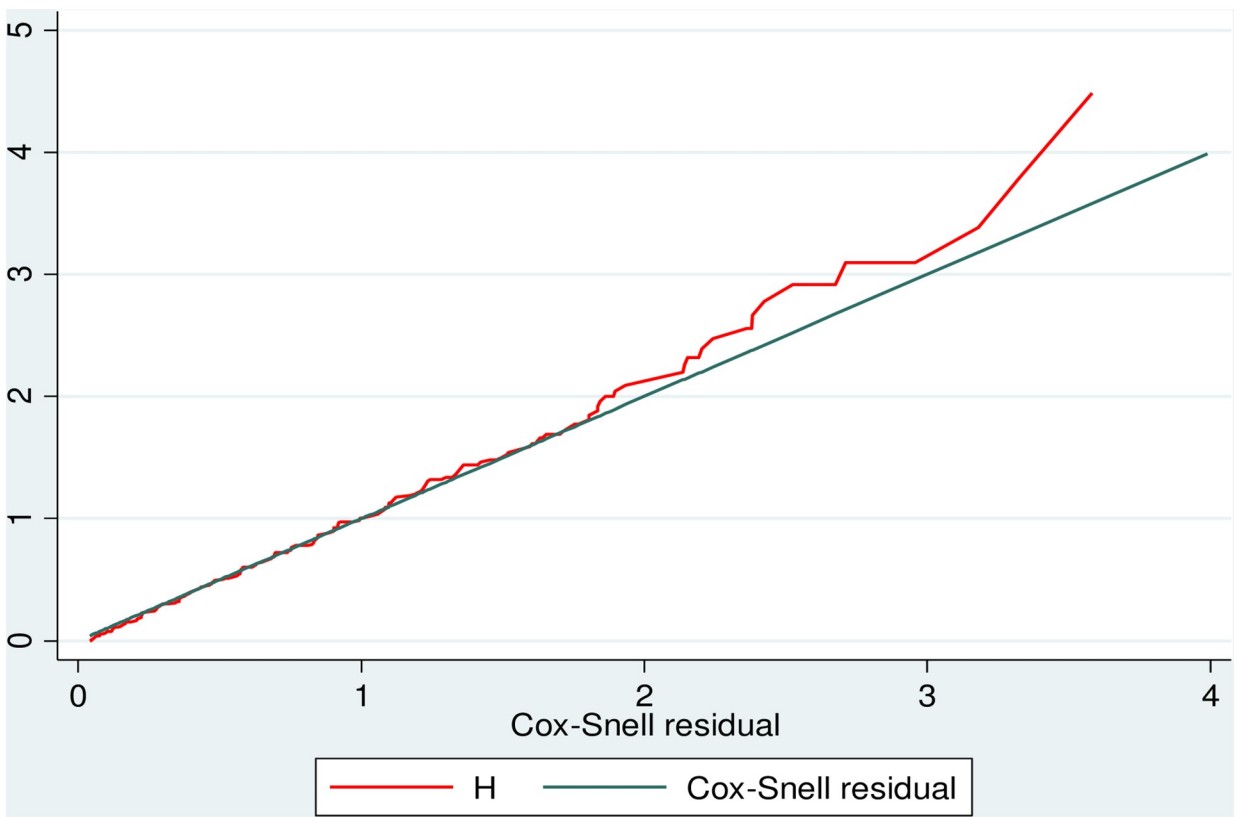

**Fig 4. The Cox-Snell residual Nelson-Aalen cumulative hazard graph on neonates with RDS admitted to the NICU at Black Lion Specialized Hospital, Ethiopia.**

parametric exponential (AIC = 987.5) and Weibull (AIC = 686.9) semi-parametric Cox-pro-portional hazard (AIC = 1123.54) models (Fig 4).

## Predictors of RD

The univariate and multivariable parametric Gompertz hazard distribution regression model was used to identify predictors of RD in neonates from admission to discharge in the NICU. Findings from the bivariate analysis showed that gestational weight, being male, having no antenatal follow-up, multiple pregnancies, neonates born via caesarean section, home delivery, PROM, maternal DM, maternal HIV/AIDS, preterm birth, neonatal sepsis, and an Apgar score of less than 7 were significantly associated with the time to discharge of neonates with RD. However, in the multivariable analysis, being male, neonates born via caesarean section, home delivery, maternal DM, preterm birth, neonatal sepsis, PROM, and an Apgar score of less than 7 were the factors which continued as statistically significant predictors of RD. The hazard ratio for RD in male neonates was 2.4 times higher than their female counterparts [AHR: 2.4 (95%CI: 1.1, 3.1)]. The current study also showed that the hazard ratio for RD among neonates born via caesarean section had nearly two times the risk compared to neonates born vaginally [AHR: 1.9 (95%CI: 1.6, 2.3)]. In this study, the risk of RD in neonates born at home was almost three times higher than those delivered at a health institution [AHR: 2.9 (95%CI: 1.5, 5,2)]. This result also indicated that neonates delivered from mothers who had DM had a 2.3 times higher risk of RD as compared with their non-DM counterparts [AHR 2.3 (95%CI: 1.4, 3.6)]. Moreover, as the gestational age increases by one week the rate of RD

**Table 3. Gompertz hazard model for predictors of RD among neonates admitted to the NICU at Black Lion Specialized Hospital, Ethiopia (N = 571).**

| Predictor | Category | RD (n, %) | Censored (%) | Total (%) | CHR (95%CI) | AHR (95%CI) |
|---|---|---|---|---|---|---|
| Mother's age (years) | <20 | 50 (20.4) | 69 (21.2) | 119 (10.9) | 1.5 (0.97, 2.4) | 1.4 (1.3, 1.9) |
| | 20–34 | 107 (43.7) | 44 (13.5) | 151 (74.6) | 1 | |
| | ≥34 | 88 (35.9) | 213 (78.4) | 301 (14.7) | 2.7 (1.18, 3.4) | 2.8(1.8, 3.3) |
| Sex | Female | 171 (48.6) | 101 (46.2.) | 272 (47.6) | 1 | |
| | Male | 181 (51.4) | 118 (53.8) | 299 (52.4) | 1.7 (1.2, 2.3) ** | 2.4 (1.1, 3.1) * |
| Place of delivery | Home | | | | 3.14 (2.3, 5.2) | 2.9 (1.5, 5.2) * |
| | Health institution | | | | 1 | |
| ANC follow up | Yes | 56 (22.8) | 113 (34.7) | 169 (29.6) | 0.4 (0.3, 0.5) ** | 0.8 (0.54, 1.19) |
| | No | 189 (77.2) | 213 (65.3) | 402 (70.4) | 1 | |
| Multiple pregnancy | Yes | 23 (9.4) | 26 (8) | 49 (8.5) | 1.6 (1.1, 2.1) ** | 1.1 (0.9, 1.6) |
| | No | 22 (90.6) | 300 (92) | 351 (91.5) | 1 | |
| PROM | Yes | 143 (58.4) | 107 (32.8) | 250 (43.8) | 1.5 (1.1, 2.0) * | 1.1(1.8, 1.5) * |
| | No | 102 (41.6) | 219 (67.2) | 321 (56.2) | 1 | |
| Mode of delivery | Caesarean section | 132 (53.8) | 103 (31.6) | 235 (41.2) | 1.6 (1.2, 2.2) ** | 1.9 (1.6, 2.3) |
| | Vaginal Caesarean section | 113 (46.2) | 223 (68.4) | 336 (58.8) | 1 | |
| HIV/AIDS | Yes | 35 (14.3) | 41 (12.6) | 76 (13.3) | 1.9 (1.3, 2.7) ** | 1.5(0.9, 2.5) |
| | No | 210 (85.7) | 285 (87.4) | 495 (86.7) | 1 | |
| Maternal DM | Yes | 39 (15.9) | 22 (6.7) | 61 (10.7) | 2.4 (1.6, 3.5) ** | 2.3 (1.4, 3.6) ** |
| | No | 206 (84.1) | 304 (93.3) | 510 (89.3) | 1 | |
| Sepsis | Yes | 189 (77.1) | 122 (37.4) | 311 (54.5) | 2.2 (1.6, 3.1) ** | 1.6 (1.1, 2.4) ** |
| | No | 56 (22.9) | 204 (62.6) | 260 (45.5) | 1 | |
| GA | <37 | 23 (13.5) | 8 | 31 (5.4) | 6.3 (3.9, 10.2) ** | 2.9 (1.6, 5.1) ** |
| | ≥37 | 61 (35.9) | 271 (67.6) | 332 (58.1) | 1 | |
| Neonatal weight (g) | <1000 | 22 (12.9) | 11 | 33 (5.8) | 3.8 (1.9, 7.5) ** | 1.9 (0.9, 4.3) |
| | 1000–1500 | 56 (32.9) | 99 (24.7) | 155 (27.1) | 1.3 (0.7, 2.4) | 0.8 (0.41, 1.6) |
| | 1500–2500 | 84 (49.4) | 257 (64.1) | 341 (59.7) | 1.1 (0.6, 1.9) | 0.8 (0.4, 1.4) |
| | ≥2500 | 8 | 34 (8.5) | 42 (7.4) | 1 | |
| First minute Apgar Score | <7 | 154 (90.6) | 283 (70.5) | 437 (76.5) | 3.2 (1.9, 5.4) * | 3.1 (1.8, 5.0) * |
| | ≥7 | 16 | 118 (29.5) | 134 (23.5) | 1 | |
| Fifth minute Apgar score | <7 | 128 (75.2) | 131 (32.7) | 259 (45.4) | 3.8 (2.7, 5.4) ** | 1.81 (1.3, 4.8) ** |
| | ≥7 | 42 (24.8) | 270 (67.3) | 312 (54.6) | 1 | |

decreased by 10% [AHR: 2.9 (95%CI: 1.6, 5.1)]. The risk of RD also increased threefold for a neonate who had an Apgar score of less than 7 as compared with one having an Apgar score greater than or equal to 7 [AHR: 3.1 (95%CI: 1.8, 5.0)]. Additionally, neonatal sepsis increases the risk of RD by 60% [AHR: 1.6 (95%CI: 1.1, 2.4)]. The last predictor for RD was to be born from mothers experiencing PROM, with neonates having a 1.11.1(1.8,1.5) times higher risk of RD than their counterparts not experiencing PROM [AHR: 1.1 (95%CI: 1.8, 1.5)] (Table 3).

## Discussion

In the current study, the overall proportion of neonates with RD admitted to the Black Lion Specialized Hospital NICU was 42.9% (95%CI: 39.3–46.1%). This finding is in line with a study conducted in the Republic of China [23]. However, our finding is higher than studies conducted in several countries including Nepal (34%) [24], India (33.4%) [10], Egypt (34.3%) [25], Pakistan (4.6%) [26], Northern Italy (20.1%) [27], and Portugal (8.83%) [28]. Variance noted in these studies may have been due to differences in the study settings which maybe

more advanced maternal newborn care services in some locations than in others. Additionally, sample size, study design, and population socio-demographic characteristics may also lead to the differences observed between studies. Interestingly, the prevalence of RD found in this study was lower than studies from Saudi Arabia (54.7%) [8], Cameroon (47.5%) [7], and Poland (54.29%) [29]. These differences may reflect the quality/qualifications of staff, public awareness to attended births, and cultural beliefs.

Based on the current finding, the overall incidence of RD was 8.1 per 100 neonate-days (95%CI: 7.3, 8.9). The common causes of RD in our study were RDS and meconium aspiration, which is similar to previous findings from Nepal and Egypt [24, 25].

This study found that there were multiple predictors of RD in neonates from Ethiopia, including preterm birth, caesarean section delivery, Apgar score < 7, sepsis, PROM, maternal DM, and home delivery. The risk of RD in male neonates was 2.4 times higher than their female counterparts; a finding which was also found in studies done in China [28] and Cameroon [7]. This aligns with the fact that male neonates have higher levels of circulating testosterone than females, which may be associated with differences in pulmonary biomechanics and vascular development. For those neonates delivered via caesarean section there was a nearly two times higher risk of developing RD than in non-caesarean births. This was also found in neonate studies from China [23], Cameroon [7], and Italy [30]. Moreover, we found that the risk of RD for neonates born at home was almost three times higher than those delivered at a health institution.

Mothers with DM bore infants 2.3 times more likely to develop RD than non-DM mothers, which is 2again a finding supported by research done in China [31]. It is possible that this relates to the fact that these neonates have plentiful glucose stores, but develop hypoglycemia because of high insulin secretion induced by maternal and fetal hyperglycemia.

Our study also found that preterm neonates had a threefold greater likelihood of RD than those who were term births, which aligns with work from Cameron [7] and Italy [30]. This finding seems to coincide with the positive association between gestational age and fetal development, thereby reducing complications as the level of prematurity is lessened. Additionally, those neonates born premature often have immature lung structures which might delay intra-pulmonary fluid absorption, a deficiency in pulmonary surfactant, and inefficient gas exchange. The risk of RD was also increased by threefold in neonates who had an Apgar score less than 7, which has been previously reported in other studies [7]. Finally, neonatal sepsis was significantly associated with the risk of developing RD as was found in Nepal and Egypt, [24, 25].

## Limitations

Since the data were collected from secondary source; some important predictors such as socio-economic factors like nutritional status of mother, educational level, birth interval might be missed which will have a significant on RD. The study area covers only TASH; its generalizability to all hospitals of the city and Ethiopia is may not be possible and this might also decrease our precision.

## Conclusion

RD was found to be a major public health problem for neonates that were admitted to NICU of Black Lion Specialized Hospital. Those neonates delivered at home, delivered through caesarean section, born preterm, who had an Apgar score < 7, or were born from diabetic mothers were most likely to develop RD. Thus, to actively reduce the risk of the development of RD in neonates, medical professionals should support pregnant mother's health wherever possible

and encourage those mothers to give birth in health care institutions, especially in premature birth situations.

## Supporting information

**S1 Text. STROBE Statement—Checklist of items that should be included in reports of cohort studies.**
(DOCX)

**S2 Text. Checklist used to assess incidence of RD and its predictors.**
(DOCX)

**S1 Table.**
(DOCX)

## Acknowledgments

We would like to thank black lion hospital neonatal ward staffs, card extractors, and data collectors whose assistance was invaluable to our completion of the study. our gratitude also goes to doctor Ryan Bell (CEO and Chief Editor Excision Editing) who have made an extensive edition in our manuscript.

## Author Contributions

**Conceptualization:** Yared Asmare Aynalem, Pammla Margaret Petrucka.

**Data curation:** Yared Asmare Aynalem, Hussien Mekonen, Tadesse Yirga Akalu, Pammla Margaret Petrucka.

**Formal analysis:** Yared Asmare Aynalem, Hussien Mekonen, Pammla Margaret Petrucka.

**Investigation:** Yared Asmare Aynalem, Hussien Mekonen.

**Methodology:** Yared Asmare Aynalem, Pammla Margaret Petrucka.

**Project administration:** Yared Asmare Aynalem, Tesfa Dejenie Habtewold.

**Resources:** Tesfa Dejenie Habtewold.

**Software:** Yared Asmare Aynalem, Tesfa Dejenie Habtewold, Aklilu Endalamaw, Wondimeneh Shibabaw Shiferaw.

**Supervision:** Yared Asmare Aynalem, Tadesse Yirga Akalu, Aklilu Endalamaw.

**Validation:** Yared Asmare Aynalem, Wondimeneh Shibabaw Shiferaw.

**Visualization:** Yared Asmare Aynalem, Wondimeneh Shibabaw Shiferaw.

**Writing – original draft:** Yared Asmare Aynalem, Aklilu Endalamaw, Wondimeneh Shibabaw Shiferaw.

**Writing – review & editing:** Yared Asmare Aynalem, Tadesse Yirga Akalu, Wondimeneh Shibabaw Shiferaw.

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
