## [Decision Letter · Decision Letter 0]

4 May 2020

PONE-D-20-01297

Incidence of respiratory distress and its predictors among neonates admitted at neonatal intensive care unit, Black Lion Specialized Hospital, Addis Ababa, Ethiopia

PLOS ONE

Dear Yared Asmare Aynalem,

Thank you for submitting your manuscript to PLOS ONE. After careful consideration, we feel that it has merit but does not fully meet PLOS ONE’s publication criteria as it currently stands. Therefore, we invite you to submit a revised version of the manuscript that addresses the points raised during the review process.

We would appreciate receiving your revised manuscript by Jun 18 2020 11:59PM. To enhance the reproducibility of your results, we recommend that if applicable you deposit your laboratory protocols in protocols.io, where a protocol can be assigned its own identifier (DOI) such that it can be cited independently in the future. For instructions see: http://journals.plos.org/plosone/s/submission-guidelines#loc-laboratory-protocols

We look forward to receiving your revised manuscript.

Kind regards,

Georg M. Schmölzer

Academic Editor

PLOS ONE

Additional Editor Comments:

Thank you for your submission,

In addition to the reviewers comments, please see below:

Please add the STROBE - Statement checklist for cohort studies when you upload the revisions.

'Ethical clearance was obtained from Addis Ababa University, College of Nursing and Midwifery Research Committee. Letters of cooperation were written to Black Lion Specialized Hospital and subsequently permission was obtained from the clinical director and relevant department and unit heads of the hospital. Following these approvals, access to the medical charts was undertaken with due diligence to maintain participant confidentiality.'

a. Please amend your current ethics statement to include the full name of the ethics committee/institutional review board(s) that approved your specific study and confirm that your named institutional review board or ethics committee specifically approved this study.

3. In your ethics statement in the manuscript and in the online submission form, please provide additional information about the patient records used in your retrospective study.

Specifically, please ensure that you have discussed whether all data were fully anonymized before you accessed them and/or whether the IRB or ethics committee waived the requirement for informed consent.

If patients provided informed written consent to have data from their medical records used in research, please include this information.

4. In your Methods section, please provide additional information about the participant recruitment method and the demographic details of your participants.

Please ensure you have provided sufficient details to replicate the analyses such as:

a) the recruitment date range (month and year), 

b) a description of how participants were recruited, and

c) descriptions of where participants were recruited and where the research took place."

6. Thank you for stating the following financial disclosure:

'The funders had no role in study design, data collection and analysis, decision to publish, or preparation of the manuscript.'

7. Please amend the manuscript submission data (via Edit Submission) to include author Pammla Margaret Petrucka.

8. Please amend your list of authors on the manuscript to ensure that each author is linked to an affiliation. Authors’ affiliations should reflect the institution where the work was done (if authors moved subsequently, you can also list the new affiliation stating “current affiliation:….” as necessary).

9. Please include your tables as part of your main manuscript and remove the individual files.

Please note that supplementary tables should be uploaded as separate "supporting information" files

10. Your ethics statement must appear in the Methods section of your manuscript. If your ethics statement is written in any section besides the Methods, please move it to the Methods section and delete it from any other section. Please also ensure that your ethics statement is included in your manuscript, as the ethics section of your online submission will not be published alongside your manuscript.

Reviewers' comments:

Reviewer's Responses to Questions

**Comments to the Author**

1. Is the manuscript technically sound, and do the data support the conclusions?

Reviewer #1: Yes

2. Has the statistical analysis been performed appropriately and rigorously? 

Reviewer #1: Yes

3. Have the authors made all data underlying the findings in their manuscript fully available?

Reviewer #1: Yes

4. Is the manuscript presented in an intelligible fashion and written in standard English?

Reviewer #1: Yes

5. Review Comments to the Author

Reviewer #1: The revised manuscript is much improved with the exception of the abstract. The topic is important. As the authors have correctly noted without knowing what causes respiratory distress in their setting you cannot develop an effective strategy to reduce the incidence and I congratulate them for this.

The abstract still needs major work. The same things are repeated 3-4 times in the abstract's result and conclusion sections and don't need to be. The first sentence of the abstract needs to be deleted as that is said in the results section. The 3rd and 4th sentences need to be revised and combined again so they are not mostly a revision of the results section. Authors might consider something like Concerned bodies Ethiopia and other low-resource nurseries with similar risk factors should develop a strategy to decrease respiratory distress in their nurseries. Based on our findings this would likely include encouraging more hospital births, better control of diabetes in pregnancy, improved neonatal resuscitation and addressing ways to decrease the need for frequent caesarean sections.

Background has been improved.

In second paragraph still have missing reference develop significant morbidity ?? source----need to add reference.

In 3rd paragraph eliminate Meconium stained amniotic fluid OR combine meconium stained amniotic fluid with meconium aspiration syndrome as that is the real cause of RD i.e. meconium stained amniotic fluid leading to meconium aspiration syndrome

Could be improved by better lead in as to why these results would be helpful and how they relate to other locales. In third paragraph of background would start paragraph off something like--Recognized causes of RD in other low and high resource countries include.......

Sentence starting with In contrast needs to be combined with 3rd paragraph and not be free standing.

In the paragraph just above table 2 the sentence half of the neonates that was delivered should read and half the neonates that were delivered via caesarean section

Tables and Graphs are approrpiate

Discussion

Would clarify the sentence beginning with such variance. I think you mean Variance noted in these studies may have been due to differences in the study settings which maybe more advanced in some locations than in others.

Would strengthen your conclusion similar to what I did in the abstract i.e. how do the results of this study inform what you and others in similar nurseries need to do to decrease the incidence of respiratory distress.

6. PLOS authors have the option to publish the peer review history of their article (what does this mean?). If published, this will include your full peer review and any attached files.

Reviewer #1: No

---

## [Author Response · Author response to Decision Letter 0]

5 Jun 2020

Dear Academic Editor!

PLOS ONE 

Response to Academic Editor and Reviewers

I am pleased to resubmit for publication version of “Incidence of respiratory distress and its predictors among neonates admitted to the neonatal intensive care unit, Black Lion Specialized Hospital, Addis Ababa, Ethiopia” for a review as original research in PLOS ONE.

The comments of the editor and the reviewers were highly insightful and enabled us to greatly improve the quality of our manuscript. Therefore, based on the editor’s and the reviewers’ concerns we have made extensive edition in our manuscript. The comments of the editor and the reviewers were highly insightful and enabled us to greatly improve the quality of our manuscript. Therefore, based on the editor’s and the reviewers’ concerns we have made extensive edition in our manuscript. Especially we have extensively edited the manuscript by a professional language editor, at Excision Editing (a fluent, native Australian, English-language speaker thoroughly edited the manuscript for language usage, spelling, and grammar) before submitting the revised version. The formatting of the text and document (text sizes and grammatical errors) were also edited. His name is called Dr. Ryan Bell(CEO and Chief Editor Excision Editing)

 In the following pages, we have addressed yours’ concerns in a point by point format. 

 We look forward to hearing from you at your earliest convenience. 

Thank you for your consideration of this manuscript! 

Kind regards,

Yared Asmare Aynalem.

On behalf of authors

Editors comment 

PONE-D-20-01297

Incidence of respiratory distress and its predictors among neonates admitted at neonatal intensive care unit, Black Lion Specialized Hospital, Addis Ababa, Ethiopia

PLOS ONE

Dear Yared Asmare Aynalem,

Thank you for submitting your manuscript to PLOS ONE. After careful consideration, we feel that it has merit but does not fully meet PLOS ONE’s publication criteria as it currently stands. Therefore, we invite you to submit a revised version of the manuscript that addresses the points raised during the review process.

Response: thank you very much for consideration.

We would appreciate receiving your revised manuscript by Jun 18 2020 11:59PM. To enhance the reproducibility of your results, we recommend that if applicable you deposit your laboratory protocols in protocols.io, where a protocol can be assigned its own identifier (DOI) such that it can be cited independently in the future. For instructions see: http://journals.plos.org/plosone/s/submission-guidelines#loc-laboratory-protocols

• A rebuttal letter that responds to each point raised by the academic editor and reviewer(s). This letter should be uploaded as separate file and labeled 'Response to Reviewers'.

• A marked-up copy of your manuscript that highlights changes made to the original version. This file should be uploaded as separate file and labeled 'Revised Manuscript with Track Changes'.

• An unmarked version of your revised paper without tracked changes. This file should be uploaded as separate file and labeled 'Manuscript'.

Response: thank you very much. We have sent the revised manuscript as based on the plose one guideline and as per editors comment 

We look forward to receiving your revised manuscript.

Kind regards,

Georg M. Schmölzer

Academic Editor

PLOS ONE

Response: thank you very much

additional Editor Comments:

Thank you for your submission,

Response: thank you very much

In addition to the reviewers comments, please see below:

Please add the STROBE - Statement checklist for cohort studies when you upload the revisions.

 Response: thank you very much for reminding us to include the STROBE statement checklist.as per your suggestion we have include it.

 Response: thank you very much

Response: thank you .we have correct our manuscript based one PLOS ONE's style requirements including the file naming. We have correct all the intended correction based on the line that the editor provided to us(plose one submission guideline) 

'Ethical clearance was obtained from Addis Ababa University, College of Nursing and Midwifery Research Committee. The full name of the ethical committee/institutional review board(s) that approved our specific study is “ College of health science ethical review bord ,Addis Ababa Universty”

Letters of cooperation were written to Black Lion Specialized Hospital and subsequently permission was obtained from the clinical director and relevant department and unit heads of the hospital. Following these approvals, access to the medical charts was undertaken with due diligence to maintain participant confidentiality.'

Response: thank you for acknowledging that 

a. Please amend your current ethics statement to include the full name of the ethics committee/institutional review board(s) that approved your specific study and confirm that your named institutional review board or ethics committee specifically approved this study.

from Addis Ababa University, College of Nursing and Midwifery Research Committee. Letters 

Response: thank you this concern .The full name of the ethical committee/institutional review board(s) that approved our specific study is “ College of health science ethical review bord ,Addis Ababa Universty”

Response: thank you .we have edited that

 3. In your ethics statement in the manuscript and in the online submission form, please provide additional information about the patient records used in your retrospective study.

Specifically, please ensure that you have discussed whether all data were fully anonymized before you accessed them and/or whether the IRB or ethics committee waived the requirement for informed consent.

If patients provided informed written consent to have data from their medical records used in research, please include this information.

 Response: we acknowledge for the issue. since the data were taken from the neonatal record, what we have done is that providing the ethical review bord letters to the hospital manager, particularly to the pediatric ward directors. then they provide to us a later permission. then we try to retrieve their data by using their card number. since it didn’t have a direct effect/harm on the neonate because of chart review. the patient was not able to gave the written informed consent. because we have reviewed the last 5 years chart of their new born. rather we try to keep the confidentiality issue by coding without listing their name.

4. In your Methods section, please provide additional information about the participant recruitment method and the demographic details of your participants.

Response: thank you very much. the demographic detail of the participants is stated as follows; study was conducted in Addis Ababa, a capital city of Ethiopia at black lion hospital. Addis Ababa has ten sub-cities in which the City lies at an altitude of 7,546 feet (2,300metres). It has twelve governmental and nine nongovernmental hospitals. The NICU of black lion hospital ward is able to accommodate a maximum of 60 patients with average of 20-40 patient’s daily admission. There are on average 5000-6000 annual admissions of neonates and 75% of admissions are from referral of different birth centers. The study was conducted from March to April 1, 2018.

Please ensure you have provided sufficient details to replicate the analyses such as:

a) the recruitment date range (month and year), 

Response: thank you very much. the recruitment date range were from January 2013 to the last of March 2018

b) a description of how participants were recruited, and

Response: thank you very much .The neonatal chart number were taken from the HMIS- data base. The total patient admitted to NICU from January 2013-last of March 2018 were 5000. We have found the number of admissions for each year. The sample were proportionally allocated for each year, and with systematics sampling; the study participants of each year were selected as follows. First, numbering the units of each year on the frame from 1 to N (N=total admission of each year), then we determine the sampling interval(K) by dividing the number of units in the population by the desired sample size of each year (n=sample size of each year) which gives 8. Then number between one and 8 at random was selected (2 were selected). This number is called the random start and the first number included in the sample. Then later Selection was conducted every 8th unit after that first number.

c) descriptions of where participants were recruited and where the research took place."

Response: thank you .The study were conducted in NICU of Black Lion hospital, the larger and the most known referal hospital of Ethiopia ,located in the capital of Ethiopia ,Addis abba .

 Response: thank you.Sorry for the inconivenec.it is to mean all data are accessible on online.

Response: thank you.all data can be accessed on online 

upload the minimal anonymized data set necessary to replicate your study findings as either Supporting Information file 

Response: thank you We have upload the minimal anonymized data set necessary to replicate our study findings as a either Supporting Information file. upload the minimal anonymized data set necessary to replicate your study findings as either Supporting Information file

6. Thank you for stating the following financial disclosure:

Response: thank you for acknowledging that 

'The funders had no role in study design, data collection and analysis, decision to publish, or preparation of the manuscript.'

a. Please clarify the sources of funding (financial or material support) for your study. List the grants or organizations that supported your study, including funding received from your institution.

Response: thank you for response.it was self-sponsored /we the authors were the source of the funding . The authors have also confirmed that no financial funding was received for the study, authorship, and publication of this article

Response: thank you. we haven’t any source of the fund. We all the authors participate in select the study design, data collection and analysis, decision to publish, or preparation of the manuscript.

Response: thank you. None of the authors received a salary .because it were self-sponsored 

d. If you did not receive any funding for this study, please state: “The authors Response: thank you .The authors received no specific funding for this work.”

Response: thank you.we included it to the cover latter 

7. Please amend the manuscript submission data (via Edit Submission) to include author Pammla Margaret Petrucka.

Response: thank you.we have included author Pammla Margaret Petrucka in the manuscript submission data via Edit Submission

8. Please amend your list of authors on the manuscript to ensure that each author is linked to an affiliation. Authors’ affiliations should reflect the institution where the work was done (if authors moved subsequently, you can also list the new affiliation stating “current affiliation:….” as necessary).

Response: thank you. We have amended the list of authors on the manuscript that each author is linked to an affiliation. But there is no any new affiliation stating “current affiliation.

9. Please include your tables as part of your main manuscript and remove the individual files.

Response: thank you. We included the table as a part of the main manuscript 

Please note that supplementary tables should be uploaded as separate "supporting information" files

Response: thank you.we have attached supplementary tables as a separate "supporting information" files

10. Your ethics statement must appear in the Methods section of your manuscript. If your ethics statement is written in any section besides the Methods, please move it to the Methods section and delete it from any other section. Please also ensure that your ethics statement is included in your manuscript, as the ethics section of your online submission will not be published alongside your manuscript.

Response: thank you.we have moved the ethical statement in the Methods section of our manuscript and we have incorporating it in the manuscript.

Reviewers' comments:

Reviewer's Responses to Questions

Comments to the Author

1. Is the manuscript technically sound, and do the data support the conclusions?

Reviewer #1: Yes

Response: thank you for acknowledgment 

2. Has the statistical analysis been performed appropriately and rigorously?

Reviewer #1: Yes

 Response: thank you for acknowledgment 

3. Have the authors made all data underlying the findings in their manuscript fully available?

Reviewer #1: Yes

 Response: thank you for acknowledgment 

4. Is the manuscript presented in an intelligible fashion and written in standard English?

Reviewer #1: Yes

 Response: thank you for acknowledgment 

5. Review Comments to the Author

Response: thank you .we took our time to correct all the concern of the reviewers 

Reviewer #1: The revised manuscript is much improved with the exception of the abstract. The topic is important. As the authors have correctly noted without knowing what causes respiratory distress in their setting you cannot develop an effective strategy to reduce the incidence and I congratulate them for this.

Response: thank you for acknowledgment 

The abstract still needs major work. The same things are repeated 3-4 times in the abstract's result and conclusion sections and don't need to be. The first sentence of the abstract needs to be deleted as that is said in the results section. The 3rd and 4th sentences need to be revised and combined again so they are not mostly a revision of the results section. Authors might consider something like Concerned bodies Ethiopia and other low-resource nurseries with similar risk factors should develop a strategy to decrease respiratory distress in their nurseries. Based on our findings this would likely include encouraging more hospital births, better control of diabetes in pregnancy, improved neonatal resuscitation and addressing ways to decrease the need for frequent caesarean sections.

Response: thank you for your concern. We have included all your contractive feedback to our papers. See the highlight. Additionally, a professional language editor (Dr. Ryan Bell CEO and Chief Editor Excision Editing)was edited the paper for the second round. 

Background has been improved.

In second paragraph still have missing reference develop significant morbidity ?? source----need to add reference.

Response: thank you for your concern. We would like to excuse for the inconvenience. we have cited it and highlighted 

In 3rd paragraph eliminate Meconium stained amniotic fluid OR combine meconium stained amniotic fluid with meconium aspiration syndrome as that is the real cause of RD i.e. meconium stained amniotic fluid leading to meconium aspiration syndrome

Response: thank you for your concern. we correct it as meconium aspiration syndrome per your recommendation. thanks once again 

Could be improved by better lead in as to why these results would be helpful and how they relate to other locales. In third paragraph of background would start paragraph off something like--Recognized causes of RD in other low and high resource countries include.......

Response: thank you for your concern.we have modified it. Also see the highlights 

Sentence starting with In contrast needs to be combined with 3rd paragraph and not be free standing.

Response: thank you for your critical revision. We have combined it as per your request and contractive feedback 

In the paragraph just above table 2 the sentence half of the neonates that was delivered should read and half the neonates that were delivered via caesarean section

Response: thank you for your critical revision. we edited it.

Tables and Graphs are appropriate

Response: thank you for acknowledgment 

Discussion

Would clarify the sentence beginning with such variance. I think you mean Variance noted in these studies may have been due to differences in the study settings which maybe more advanced in some locations than in others.

Response: thank you for your critical revision.we have include the suggestion given by our reviewer.

Would strengthen your conclusion similar to what I did in the abstract i.e. how do the results of this study inform what you and others in similar nurseries need to do to decrease the incidence of respiratory distress.

 Response: We thank you very much for this important recommendation. Based on your recommendations, we rearranged it 

6. PLOS authors have the option to publish the peer review history of their article (what does this mean?). If published, this will include your full peer review and any attached files.

Do you want your identity to be public for this peer review? For information about this choice, including consent withdrawal, please see our Privacy Policy.

Reviewer #1: No

Response: We thank you for reminding us to publish peer review history of their article. sorry for the inconvenience. we the authors are happy to published it.

While revising your submission, please upload your figure files to the Preflight Analysis and Conversion Engine (PACE) digital diagnostic tool, https://pacev2.apexcovantage.com/. PACE helps ensure that figures meet PLOS requirements. To use PACE, you must first register as a user. Registration is free. Then, login and navigate to the UPLOAD tab, where you will find detailed instructions on how to use the tool. If you encounter any issues or have any questions when using PACE, please email us at figures@plos.org. Please note that Supporting Information files do not need this s

Response: We thank you for reminding us to use PACE for uploading our figures. We have attached the figures with PACE as per your recommendation 

Thank you

---

## [Decision Letter · Decision Letter 1]

16 Jun 2020

PONE-D-20-01297R1

Incidence of respiratory distress and its predictors among neonates admitted to the neonatal intensive care unit, Black Lion Specialized Hospital, Addis Ababa, Ethiopia

PLOS ONE

Dear Dr. Yared Asmare Aynalem,

Thank you for submitting your manuscript to PLOS ONE. After careful consideration, we feel that it has merit but does not fully meet PLOS ONE’s publication criteria as it currently stands. Therefore, we invite you to submit a revised version of the manuscript that addresses the points raised during the review process.

We look forward to receiving your revised manuscript.

Kind regards,

Georg M. Schmölzer

Academic Editor

PLOS ONE

Reviewers' comments:

Reviewer's Responses to Questions

**Comments to the Author**

1. If the authors have adequately addressed your comments raised in a previous round of review and you feel that this manuscript is now acceptable for publication, you may indicate that here to bypass the “Comments to the Author” section, enter your conflict of interest statement in the “Confidential to Editor” section, and submit your "Accept" recommendation.

Reviewer #1: All comments have been addressed

2. Is the manuscript technically sound, and do the data support the conclusions?

Reviewer #1: Yes

3. Has the statistical analysis been performed appropriately and rigorously? 

Reviewer #1: Yes

4. Have the authors made all data underlying the findings in their manuscript fully available?

Reviewer #1: Yes

5. Is the manuscript presented in an intelligible fashion and written in standard English?

Reviewer #1: Yes

6. Review Comments to the Author

Reviewer #1: Under Background

Probably more correct to say. “It is often seen during the transition from fetal to neonatal life”. Not triggered by

7. PLOS authors have the option to publish the peer review history of their article (what does this mean?). If published, this will include your full peer review and any attached files.

Reviewer #1: No

---

## [Author Response · Author response to Decision Letter 1]

16 Jun 2020

Dear Academic Editor!

PLOS ONE 

 A letter Accompanying Revision in Response to editor and Reviewers Comment

 I am pleased to resubmit for publication version of “Incidence of respiratory distress and its predictors among neonates admitted to the neonatal intensive care unit, Black Lion Specialized Hospital, Addis Ababa, Ethiopia” for a review as original research in PLOS ONE.

financial disclosure: The authors confirmed that no financial funding was received for the study. None of the authors received specific funding for this work. 

 The comments of the editor and reviewers were highly insightful and enabled us to greatly improve the quality of our manuscript. Therefore, based on the editor and reviewers’ concerns we have made extensive edition in our manuscript. In the following pages, we have addressed yours’ concerns in a point by point format. 

 We look forward to hearing from you at your earliest convenience. 

 Thank you for your consideration of this manuscript! 

 Sincerely, 

 Yared Asmare Aynalem

 On behalf of authors

PONE-D-20-01297R1

Incidence of respiratory distress and its predictors among neonates admitted to the neonatal intensive care unit, Black Lion Specialized Hospital, Addis Ababa, Ethiopia

PLOS ONE

Dear Dr. Yared Asmare Aynalem,

Thank you for submitting your manuscript to PLOS ONE. After careful consideration, we feel that it has merit but does not fully meet PLOS ONE’s publication criteria as it currently stands. Therefore, we invite you to submit a revised version of the manuscript that addresses the points raised during the review process.

Response: We thank you very much for consideration

Response: We thank you very much. We have submitted Response to Reviewers, Revised Manuscript with Track Changes and Manuscript separately 

Response: We thank you. We have included the financial disclosure as The authors have also confirmed that no financial funding was received for the study, authorship, and publication of this article.

the figures were uploaded with the plos one guide line 

Response. Thank you. We think that it is not applicable 

We look forward to receiving your revised manuscript.

Kind regards,

Georg M. Schmölzer

Academic Editor

PLOS ONE

Response. Big thanks. We will send it as soon as possible

Response

Reviewer #

Reviewers' comments:

Reviewer's Responses to Questions

Comments to the Author

1. If the authors have adequately addressed your comments raised in a previous round of review and you feel that this manuscript is now acceptable for publication, you may indicate that here to bypass the “Comments to the Author” section, enter your conflict of interest statement in the “Confidential to Editor” section, and submit your "Accept" recommendation.

Reviewer #1: All comments have been addressed

Response: thank you very much for acknowledging it

2. Is the manuscript technically sound, and do the data support the conclusions?

Reviewer #1: Yes

 Response: thank you very much.

3. Has the statistical analysis been performed appropriately and rigorously?

Reviewer #1: Yes

 Response: thank you very much.

4. Have the authors made all data underlying the findings in their manuscript fully available?

Reviewer #1: Yes

 Response: thank you very much.

5. Is the manuscript presented in an intelligible fashion and written in standard English?

Reviewer #1: Yes

 Response: thank you very much.

6. Review Comments to the Author

Reviewer #1: Under Background

Probably more correct to say. “It is often seen during the transition from fetal to neonatal life”. Not triggered by

Response: thank you very much. Response: Based on your recommendations, we have addressed the issue. See the track change

7. PLOS authors have the option to publish the peer review history of their article (what does this mean?). If published, this will include your full peer review and any attached files.

Do you want your identity to be public for this peer review? For information about this choice, including consent withdrawal, please see our Privacy Policy.

Reviewer #1: No

Response: Thank you

 Response: thank you. We have uploaded figure files to the Preflight Analysis and Conversion Engine (PACE) digital diagnostic tool.

thank you

---

## [Editor Report · Decision Letter 2]

18 Jun 2020

Incidence of respiratory distress and its predictors among neonates admitted to the neonatal intensive care unit, Black Lion Specialized Hospital, Addis Ababa, Ethiopia

PONE-D-20-01297R2

Dear Dr. Yared Asmare Aynalem,

We’re pleased to inform you that your manuscript has been judged scientifically suitable for publication and will be formally accepted for publication once it meets all outstanding technical requirements.

Kind regards,

Georg M. Schmölzer

Academic Editor

PLOS ONE
---

## [Editor Report · Acceptance letter]

22 Jun 2020

PONE-D-20-01297R2 

 Incidence of respiratory distress and its predictors among neonates admitted to the neonatal intensive care unit, Black Lion Specialized Hospital, Addis Ababa, Ethiopia

Dear Dr. Aynalem:

I'm pleased to inform you that your manuscript has been deemed suitable for publication in PLOS ONE. Congratulations! Your manuscript is now with our production department. 

Kind regards, 

on behalf of

Dr. Georg M. Schmölzer 

Academic Editor

PLOS ONE